# Towards Understanding the Importance of Shortcut Connections in Residual Networks

**Tianyi Liu**[*]  
Georgia Tech

**Minshuo Chen**[*]  
Georgia Tech

**Mo Zhou**  
Duke University

**Simon S. Du**  
Institute for Advanced Study

**Enlu Zhou**  
Georgia Tech

**Tuo Zhao**  
Georgia Tech

## Abstract

Residual Network (ResNet) is undoubtedly a milestone in deep learning. ResNet is equipped with shortcut connections between layers, and exhibits efficient training using simple first order algorithms. Despite of the great empirical success, the reason behind is far from being well understood. In this paper, we study a two-layer non-overlapping convolutional ResNet. Training such a network requires solving a non-convex optimization problem with a spurious local optimum. We show, however, that gradient descent combined with proper normalization, avoids being trapped by the spurious local optimum, and converges to a global optimum in polynomial time, when the weight of the first layer is initialized at $0$, and that of the second layer is initialized arbitrarily in a ball. Numerical experiments are provided to support our theory.

## 1 Introduction

Neural Networks have revolutionized a variety of real world applications in the past few years, such as computer vision (Krizhevsky et al., 2012; Goodfellow et al., 2014; Long et al., 2015), natural language processing (Graves et al., 2013; Bahdanau et al., 2014; Young et al., 2018), etc. Among different types of networks, Residual Network (ResNet, He et al. (2016a)) is undoubted a milestone. ResNet is equipped with shortcut connections, which skip layers in the forward step of an input. Similar idea also appears in the Highway Networks (Srivastava et al., 2015), and further inspires densely connected convolutional networks (Huang et al., 2017).

ResNet owes its great success to a surprisingly efficient training compared to the widely used feedforward Convolutional Neural Networks (CNN, Krizhevsky et al. (2012)). Feedforward CNNs are seldomly used with more than 30 layers in the existing literature. There are experimental results suggest that very deep feedforward CNNs are significantly slow to train, and yield worse performance than their shallow counterparts (He et al., 2016a). However, simple first order algorithms such as stochastic gradient descent and its variants are able to train ResNet with hundreds of layers, and achieve better performance than the state-of-the-art. For example, ResNet-152 (He et al., 2016a), consisting of 152 layers, achieves a $19.38\%$ top-1 error on ImageNet. He et al. (2016b) also demonstrated a more aggressive ResNet-1001 on the CIFAR-10 data set with 1000 layers. It achieves a $4.92\%$ error — better than shallower ResNets such as ResNet-110.

Despite the great success and popularity of ResNet, the reason why it can be efficiently trained is still largely unknown. One line of research empirically studies ResNet and provides intriguing observations. Veit et al. (2016), for example, suggest that ResNet can be viewed as a collection of weakly dependent smaller networks of varying sizes. More interestingly, they reveal that these

---

[*]Equal contribution.

smaller networks alleviate the vanishing gradient problem. Balduzzi et al. (2017) further elaborate on the vanishing gradient problem. They show that the gradient in ResNet only decays sublinearly in contrast to the exponential decay in feedforward neural networks. Recently, Li et al. (2018) visualize the landscape of neural networks, and show that the shortcut connection yields a smoother optimization landscape. In spite of these empirical evidences, rigorous theoretical justifications are seriously lacking.

Another line of research theoretically investigates ResNet with simple network architectures. Hardt and Ma (2016) show that linear ResNet has no spurious local optima (local optima that yield larger objective values than the global optima). Later, Li and Yuan (2017) study using Stochastic Gradient Descent (SGD) to train a two-layer ResNet with only one unknown layer. They show that the optimization landscape has no spurious local optima and saddle points. They also characterize the local convergence of SGD around the global optimum. These results, however, are often considered to be overoptimistic, due to the oversimplified assumptions.

To better understand ResNet, we study a two-layer non-overlapping convolutional neural network, whose optimization landscape contains a spurious local optimum. Such a network was first studied in Du et al. (2017). Specifically, we consider

$$g(v, a, Z) = a^\top \sigma \left( Z^\top v \right), \tag{1}$$

where $Z \in \mathbb{R}^{p \times k}$ is an input, $a \in \mathbb{R}^k, v \in \mathbb{R}^p$ are the output weight and the convolutional weight, respectively, and $\sigma$ is the element-wise ReLU activation. Since the ReLU activation is positive homogeneous, the weights $a$ and $v$ can arbitrarily scale with each other. Thus, we impose the assumption $\|v\|_2 = 1$ to make the neural network identifiable. We further decompose $v = \mathbb{1}/\sqrt{p} + w$ with $\mathbb{1}$ being a vector of 1's in $\mathbb{R}^p$, and rewrite (1) as

$$f(w, a, Z) = a^\top \sigma \left( Z^\top (\mathbb{1}/\sqrt{p} + w) \right), \tag{2}$$

Here $\mathbb{1}/\sqrt{p}$ represents the average pooling shortcut connection, which allows a direct interaction between the input $Z$ and the output weight $a$.

We investigate the convergence of training ResNet by considering a realizable case. Specifically, the training data is generated from a teacher network with true parameters $a^*, v^*$ with $\|v^*\|_2 = 1$. We aim to recover the teacher neural network using a student network defined in (2) by solving an optimization problem:

$$(\widehat{w}, \widehat{a}) = \underset{w, a}{\operatorname{argmin}} \frac{1}{2} \mathbb{E}_Z \left[ f(w, a, Z) - g(v^*, a^*, Z) \right]^2, \tag{3}$$

where $Z$ is independent Gaussian input. Although largely simplified, (3) is nonconvex and possesses a nuisance — There exists a spurious local optimum (see an explicit characterization in Section 2). Early work, Du et al. (2017), show that when the student network has the same architecture as the teacher network, GD with random initialization can be trapped in a spurious local optimum with a constant probability[2]. A natural question here is

### Does the shortcut connection ease the training?

This paper suggests a positive answer: When initialized with $w = 0$ and $a$ arbitrarily in a ball, GD with proper normalization converges to a global optimum of (3) in polynomial time, under the assumption that $(v^*)^\top (\mathbb{1}/\sqrt{p})$ is close to 1. Such an assumption requires that there exists a $w^*$ of relatively small magnitude, such that $v^* = \mathbb{1}/\sqrt{p} + w^*$. This assumption is supported by both empirical and theoretical evidences. Specifically, the experiments in Li et al. (2016) and Yu et al. (2018), show that the weight in well-trained deep ResNet has a small magnitude, and the weight for each layer has vanishing norm as the depth tends to infinity. Hardt and Ma (2016) suggest that, when using linear ResNet to approximate linear transformations, the norm of the weight in each layer scales as $O(1/D)$ with $D$ being the depth. Bartlett et al. (2018) further show that deep nonlinear ResNet, with the norm of the weight of order $O(\log D/D)$, is sufficient to express differentiable functions under certain regularity conditions. These results motivate us to assume $w^*$ is relatively small.

Our analysis shows that the convergence of GD exhibits 2 stages. Specifically, our initialization guarantees $w$ is sufficiently away from the spurious local optimum. In the first stage, with proper step

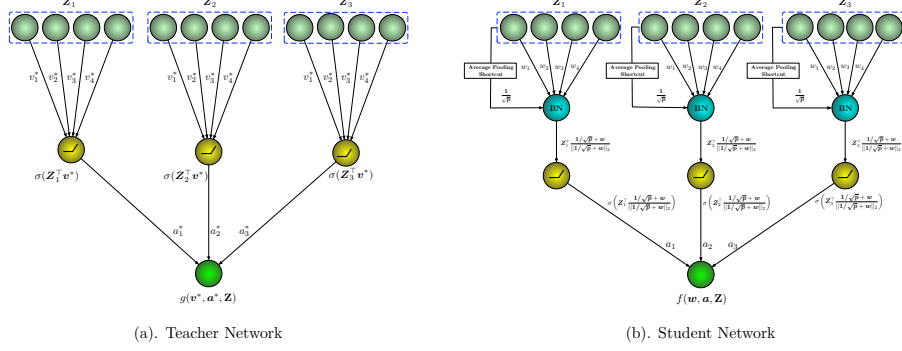

(a). Teacher Network                              (b). Student Network

Figure 2: Illustrative examples of the teacher and student networks with $k = 3$ and $p = 4$. BN notes batch normalization.

sizes, we show that the shortcut connection helps the algorithm avoid being attracted by the spurious local optima. Meanwhile, the shortcut connection guides the algorithm to evolve towards a global optimum. In the second stage, the algorithm enters the basin of attraction of the global optimum. With properly chosen step sizes, $w$ and $a$ jointly converge to the global optimum.

Our analysis thus explains why ResNet benefits training, when the weights are simply initialized at zero (Li et al., 2016), or using the Fixup initialization in Zhang et al. (2019). We remark that our choice of step sizes is also related to learning rate warmup (Goyal et al., 2017), and other learning rate schemes for more efficient training of neural networks (Smith, 2017; Smith and Topin, 2018). We refer readers to Section 5 for a more detailed discussion.

**Notations:** Given a vector $v = (v_1, \ldots, v_m)^\top \in \mathbb{R}^m$, we denote the Euclidean norm $\|v\|_2^2 = v^\top v$. Given two vectors $u, v \in \mathbb{R}^d$, we denote the angle between them as $\angle(u, v) = \arccos \frac{u^\top v}{\|u\|_2 \|v\|_2}$, and the inner product as $\langle u, v \rangle = u^\top v$. We denote $\mathbb{1} \in \mathbb{R}^d$ as the vector of all the entries being 1. We also denote $\mathbb{B}_0(r) \in \mathbb{R}^d$ as the Euclidean ball centered at 0 with radius $r$.

## 2 Model and Algorithm

**Model.** We consider the realizable setting where the label is generated from a noiseless teacher network in the following form

$$g(v^*, a^*, Z) = \sum_{j=1}^{k} a_j^* \sigma\left(Z_j^\top v^*\right). \tag{4}$$

Here $v^*, a^*, Z_j$'s are the true convolutional weight, true output weight, and input. $\sigma$ denotes the element-wise ReLU activation. Our student network is defined in (2). For notational convenience, we expand the second layer and rewrite (2) as

$$f(w, a, Z) = \sum_{j=1}^{k} a_j \sigma\left(Z_j^\top (\mathbb{1}/\sqrt{p} + w)\right), \tag{5}$$

where $w \in \mathbb{R}^p$, $a_j \in \mathbb{R}$, and $Z_j \in \mathbb{R}^p$ for all $j = 1, 2, \ldots, k$. We assume the input data $Z_j$'s are identically independently sampled from $\mathcal{N}(0, I)$. Note that the above network is not identifiable, because of the positive homogeneity of the ReLU function, that is $\mathbb{1}\sqrt{p} + w$ and $a$ can scale with each other by any positive constant without changing the output value. Thus, to achieve identifiability, instead of (5), we propose to train the following student network,

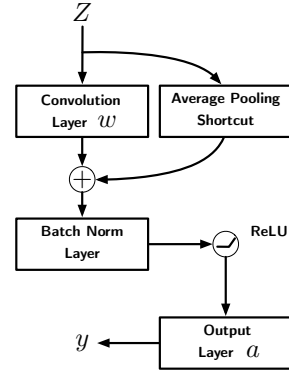

Figure 1: The non-overlapping two layer residual network with normalization layer.

$$f(w, a, Z) = \sum_{j=1}^{k} a_j \sigma\left(Z_j^\top \frac{\mathbb{1}/\sqrt{p} + w}{\|\mathbb{1}/\sqrt{p} + w\|_2}\right). \tag{6}$$

An illustration of (6) is provided in Figure 1. An example of the teacher network (4) and the student network (6) is shown in Figure 2.

We then recover $(v^*, a^*)$ of our teacher network by solving a nonconvex optimization problem

$$\min_{w, a} \mathcal{L}(w, a) = \frac{1}{2} \mathbb{E}_Z[g(v^*, a^*, Z) - f(w, a, Z)]^2. \tag{7}$$

Recall that we assume $\|v^*\|_2 = 1$. One can easily verify that (7) has global optima and spurious local optima. The characterization is analogous to Du et al. (2017), although the objective is different.

**Proposition 1.** *For any constant $\alpha > 0$, $(w, a)$ is a global optimum of (7), if $\mathbb{1}/\sqrt{p} + w = \alpha v^*$ and $a = a^*$; $(w, a)$ is a spurious local optimum of (7), if $\mathbb{1}/\sqrt{p} + w = -\alpha v^*$ and $a = (\mathbb{1}\mathbb{1}^\top + (\pi - 1)I)^{-1}(\mathbb{1}\mathbb{1}^\top - I)a^*$.*

The proof is adapted from Du et al. (2017), and the details are provided in Appendix B.1.

Now we formalize the assumption on $v^*$ in Section 1, which is supported by the theoretical and empirical evidence in Li et al. (2016); Yu et al. (2018); Hardt and Ma (2016); Bartlett et al. (2018).

**Assumption 1** (Shortcut Prior). *There exists a $w^*$ with $\|w^*\|_2 \leq 1$, such that $v^* = w^* + \mathbb{1}/\sqrt{p}$.*

Assumption 1 implies $(\mathbb{1}/\sqrt{p})^\top v^* \geq 1/2$. We remark that our analysis actually applies to any $w^*$ satisfying $\|w^*\|_2 \leq c$ for any positive constant $c \in (0, \sqrt{2})$. Here we consider $\|w^*\|_2 \leq 1$ to ease the presentation. Throughout the rest of the paper, we assume this assumption holds true.

**GD with Normalization.** We solve the optimization problem (7) by gradient descent. Specifically, at the $(t + 1)$-th iteration, we compute

$$
\begin{aligned}
\widetilde{w}_{t+1} &= w_t - \eta_w \nabla_w \mathcal{L}(w_t, a_t), \\
w_{t+1} &= \frac{\mathbb{1}/\sqrt{p} + \widetilde{w}_{t+1}}{\|\mathbb{1}/\sqrt{p} + \widetilde{w}_{t+1}\|_2} - \frac{\mathbb{1}}{\sqrt{p}}, \\
a_{t+1} &= a_t - \eta_a \nabla_a \mathcal{L}(w_t, a_t).
\end{aligned}
\tag{8}
$$

Note that we normalize $\mathbb{1}/\sqrt{p} + w$ in (8), which essentially guarantees $\mathrm{Var}\left(Z_j^\top (\mathbb{1}/\sqrt{p} + w_{t+1})\right) = 1$. As $Z_j$ is sampled from $N(0, I)$, we further have $\mathbb{E}\left(Z_j^\top (\mathbb{1}/\sqrt{p} + w_{t+1})\right) = 0$. The normalization step in (8) can be viewed as a population version of the widely used batch normalization trick to accelerate the training of neural networks (Ioffe and Szegedy, 2015). Moreover, (7) has one unique optimal solution under such a normalization. Specifically, $(w^*, a^*)$ is the unique global optimum, and $(\bar{w}, \bar{a})$ is the only spurious local optimum along the solution path, where $\bar{w} = -(\mathbb{1}/\sqrt{p}) - v^*$ and $\bar{a} = (\mathbb{1}\mathbb{1}^\top + (\pi - 1)I)^{-1}(\mathbb{1}\mathbb{1}^\top - I)a^*$.

We initialize our algorithm at $(w_0, a_0)$ satisfying: $w_0 = 0$ and $a_0 \in \mathbb{B}_0(|\mathbb{1}^\top a^*|/\sqrt{k})$. We set $a_0$ with a magnitude of $O(1/\sqrt{k})$ to match common initialization techniques (Glorot and Bengio, 2010; LeCun et al., 2012; He et al., 2015). We highlight that our algorithm starts with an arbitrary initialization on $a$, which is different from random initialization. The step sizes $\eta_a$ and $\eta_w$ will be specified later in our analysis.

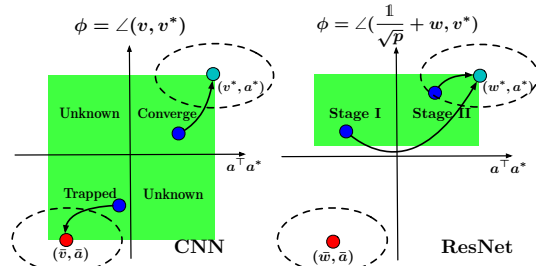

## 3 Convergence Analysis

We characterize the algorithmic behavior of the gradient descent algorithm. Our analysis shows that under Assumption 1, the convergence of GD exhibits two stages. In the first stage, the algorithm avoids being trapped by the spurious local optimum. Given the algorithm is sufficiently away from the spurious local optima, the algorithm enters the basin of attraction of the global optimum and finally converge to it.

Figure 3: The left panel shows random initialization on feedforward CNN can be trapped in the spurious local optimum with probability at least $1/4$ (Du et al., 2017). The right panel demonstrates: 1). Under the shortcut prior, our initialization of $(w, a)$ avoids starting near the spurious local optimum; 2). Convergence of GD exhibits two stages (I. improvement of $a$ and avoid being attracted by $(\bar{w}, \bar{a})$ II. joint convergence).

To present our main result, we begin with some notations. Denote

$$
\phi_t = \angle(\mathbb{1}/\sqrt{p} + w_t, \mathbb{1}/\sqrt{p} + w^*)
$$

as the angle between $\mathbb{1}/\sqrt{p} + w_t$ and the ground truth at the $t$-th iteration. Throughout the rest of the paper, we assume $\|a^*\|_2$ is a constant. The notation $\widetilde{O}(\cdot)$ hides $\mathrm{poly}(\|a^*\|_2)$, $\mathrm{poly}(\frac{1}{\|a^*\|_2})$, and $\mathrm{polylog}(\|a^*\|_2)$ factors. Then we state the convergence of GD in the following theorem.

**Theorem 2** (Main Results). *Let the GD algorithm defined in Section 2 be initialized with $w_0 = 0$ and arbitrary $a_0 \in \mathbb{B}_0(|\mathbb{1}^\top a^*|/\sqrt{k})$. Then the algorithm converges in two stages:*

**Stage I: Avoid the spurious local optimum** *(Theorem 4): We choose $\eta_a = O(1/k^2)$ and $\eta_w = \widetilde{O}(1/k^4)$. Then there exists $T_1 = \widetilde{O}(1/\eta_a)$, such that $m \le a_{T_1}^\top a^* \le M$ and $\phi_{T_1} \le \frac{5}{12}\pi$ hold for some constants $M > m > 0$.*

**Stage II: Converge to the global optimum** *(Theorem 13): After $T_1$ iterations, we restart the counter, and choose $\eta = \eta_a = \eta_w = \widetilde{O}(1/k^2)$. Then for any $\delta > 0$, any $t \ge T_2 = \widetilde{O}(\frac{1}{\eta}\log\frac{1}{\delta})$, we have $\|w_t - w^*\|_2^2 \le \delta$ and $\|a_t - a^*\|_2^2 \le 5\delta$.*

Note that the set $\{(w_t, a_t) \mid a_t^\top a^* \in [m, M], \phi_t \le 5\pi/12\}$ belongs to be the basin of attraction around the global optimum (Lemma 11), where certain regularity condition (partial dissipativity) guides the algorithm toward the global optimum. Hence, after the algorithm enters the second stage, we increase the step size $\eta_w$ of $w$ for a faster convergence. Figure 3 demonstrates the initialization of $(w, a)$, and the convergence of GD both on CNN in Du et al. (2017) and our ResNet model.

We start our convergence analysis with the definition of partial dissipativity for $\mathcal{L}$.

**Definition 3** (Partial Dissipativity). *Given any $\delta \ge 0$ and a constant $c \ge 0$, $\nabla_w \mathcal{L}$ is $(c, \delta)$-partially dissipative with respect to $w^*$ in a set $\mathcal{K}_\delta$, if for every $(w, a) \in \mathcal{K}_\delta$, we have*

$$\langle -\nabla_w \mathcal{L}(w, a), w^* - w \rangle \ge c\|w - w^*\|_2^2 - \delta;$$

*$\nabla_a \mathcal{L}$ is $(c, \delta)$-partially dissipative with respect to $a^*$ in a set $\mathcal{A}_\delta$, if for every $(w, a) \in \mathcal{A}_\delta$, we have*

$$\langle -\nabla_a \mathcal{L}(w, a), a^* - a \rangle \ge c\|a - a^*\|_2^2 - \delta.$$

*Moreover, If $\mathcal{K}_\delta \cap \mathcal{A}_\delta \ne \emptyset$, $\nabla\mathcal{L}$ is $(c, 2\delta)$-jointly dissipative with respect to $(w^*, a^*)$ in $\mathcal{K}_\delta \cap \mathcal{A}_\delta$, i.e., for every $(w, a) \in \mathcal{K}_\delta \cap \mathcal{A}_\delta$, we have*

$$\langle -\nabla_w \mathcal{L}(w, a), w^* - w \rangle + \langle -\nabla_a \mathcal{L}(w, a), a^* - a \rangle \ge c(\|w - w^*\|_2^2 + \|a - a^*\|_2^2) - 2\delta.$$

The concept of dissipativity is originally used in dynamical systems (Barrera and Jara, 2015), and is defined for general operators. It suffices to instantiate the concept to gradients here for our convergence analysis. The partial dissipativity for perturbed gradients is used in Zhou et al. (2019) to study the convergence behavior of Perturbed GD. The variational coherence studied in Zhou et al. (2017) and one point convexity studied in Li and Yuan (2017) can be viewed as special examples of partial dissipativity.

## 3.1 Stage I: Avoid the Spurious Local Optimum

We first show with properly chosen step sizes, GD algorithm can avoid being trapped by the spurious local optimum. We propose to update $w, a$ using different step sizes. We formalize our result in the following theorem.

**Theorem 4.** *Initialize with arbitrary $a_0 \in \mathbb{B}_0(|\mathbb{1}^\top a^*|/\sqrt{k})$ and $w_0 = 0$. We choose step sizes*

$$\eta_a = \frac{\pi}{20(k + \pi - 1)^2} = O\left(1/k^2\right), \quad and \quad \eta_w = C\|a^*\|_2^2 \eta_a^2 = \widetilde{O}(\eta_a^2)$$

*for some constant $C > 0$. Then, we have*

$$\phi_t \le 5\pi/12 \quad and \quad 0 \le m \le a_t^\top a^* \le M, \tag{9}$$

*for all $t \in [T_1, T]$, where*

$$T_1 = \widetilde{O}(1/\eta_a), \ \ T = O(1/\eta_a^2), \ \ m = \|a^*\|_2^2/5, \ \ and \ \ M = 3\|a^*\|_2^2 + 2\left(\mathbb{1}^\top a^*\right)^2.$$

*Proof Sketch.* Due to the space limit, we only provide a proof sketch here. The detailed proof is deferred to Appendix B.2. We prove the two arguments in (9) in order. Before that, we first show our initialization scheme guarantees an important bound on $a$, as stated in the following lemma.

**Lemma 5.** *Given $a_0 \in \mathbb{B}_0(|\mathbb{1}^\top a^*|/\sqrt{k})$, we choose $\eta_a \le \frac{2\pi}{k + \pi - 1}$. Then for any $t > 0$,*

$$-3\left(\mathbb{1}^\top a^*\right)^2 \le \mathbb{1}^\top a^* \mathbb{1}^\top a_t - \left(\mathbb{1}^\top a^*\right)^2 \le 0. \tag{10}$$

Under the shortcut prior assumption 1 that $w_0$ is close to $w^*$, the update of $w$ should be more conservative to provide enough accuracy for $a$ to make progress. Based on Lemma 5, the next lemma shows that when $\eta_w$ is small enough, $\phi_t$ stays acute ($\phi_t < \frac{\pi}{2}$), i.e., $w$ is sufficiently away from $\bar{w} = -(\mathbb{1}/\sqrt{p}) - v^*$ .

**Lemma 6.** *Given $w_0 = 0$ and $a_0 \in \mathbb{B}_0(|\mathbb{1}^\top a^*|/\sqrt{k})$, we choose $\eta_a < \frac{2\pi}{k+\pi-1}$ and $\eta_w = C\|a^*\|_2^2 \eta_a^2 = \widetilde{O}(\eta_a^2)$ for some absolute constant $C > 0$. Then for all $t \leq T = O(1/\eta_a^2)$,*

$$\phi_t \leq 5\pi/12. \tag{11}$$

We want to remark that (10) and (11) are two of the key conditions that define the partially dissipative region of $\nabla_a \mathcal{L}$ , as shown in the following lemma.

**Lemma 7.** *For any $(w, a) \in \mathcal{A}$, $\nabla_a \mathcal{L}$ satisfies*

$$\langle -\nabla_a \mathcal{L}(w, a), a^* - a \rangle \geq (1/10\pi)\|a - a^*\|_2^2, \tag{12}$$

*where $\mathcal{A} = \left\{ (w, a) \mid a^\top a^* \leq \frac{1}{20}\|a^*\|_2^2 \text{ or } \|a - a^*/2\|_2^2 \geq \|a^*\|_2^2, \|w + \mathbb{1}/\sqrt{p}\|_2 = 1, \phi \leq \frac{5}{12}\pi, -3(\mathbb{1}^\top a^*)^2 \leq \mathbb{1}^\top a^* \mathbb{1}^\top a - (\mathbb{1}^\top a^*)^2 \leq 0 \right\}$.*

Please refer to Appendix B.2.3 for a detailed proof. Note that with arbitrary initialization of $a$, $a^\top a^* \leq \frac{1}{20}\|a^*\|_2^2$ or $\|a - a^*/2\|_2^2 \geq \|a^*\|_2^2$ possibly holds at $a_0$. In this case, $(w_0, a_0)$ falls in $\mathcal{A}$, and (12) ensures the improvement of $a$.

**Lemma 8.** *Given $(w_0, a_0) \in \mathcal{A}$, we choose $\eta_a < \frac{\pi}{20(k+\pi-1)^2}$. Then there exists $\tau_{11} = O(1/\eta_a)$, such that $\frac{1}{20}\|a^*\|_2^2 \leq a_{\tau_{11}}^\top a^* \leq 2\|a^*\|_2^2$.*

One can easily verify that $a^\top a^* \leq 2\|a^*\|_2^2$ holds for any $a \in \mathbb{B}_0(|\mathbb{1}^\top a^*|/\sqrt{k})$. Together with Lemma 8, we claim that even with arbitrary initialization, the iterates can always enter the region with $a^\top a^*$ positive and bounded in polynomial time. The next lemma shows that with proper chosen step sizes, $a^\top a^*$ stays positive and bounded.

**Lemma 9.** *Suppose $\frac{1}{20}\|a^*\|_2^2 \leq a_0^\top a^* \leq 2\|a^*\|_2^2$, $\phi_t \leq \frac{5}{12}\pi$, and $-3\left(\mathbb{1}^\top a^*\right)^2 \leq \mathbb{1}^\top a^* \mathbb{1}^\top a_t - \left(\mathbb{1}^\top a^*\right)^2 \leq 0$ holds for all $t$. Choose $\eta_a < \frac{2\pi}{\pi-1}$, then we have for all $t \geq \tau_{12} = \widetilde{O}(1/\eta_a)$,*

$$\|a^*\|_2^2/5 \leq a_t^\top a^* \leq 3\|a^*\|_2^2 + 2\left(\mathbb{1}^\top a^*\right)^2 .$$

Take $T_1 = \tau_{11} + \tau_{12}$, and we complete the proof. $\qquad\qquad\square$

In Theorem 4, we choose a conservative $\eta_w$. This brings two benefits to the training process: 1). $w$ stays away from $\bar{w}$. The update on $w$ is quite limited, since $\eta_w$ is small. Hence, $w$ is kept sufficiently away from $\bar{w}$, even if $w$ moves towards $\bar{w}$ in every iteration); 2). $a$ continuously updates toward $a^*$.

Theorem 4 ensures that under the shortcut prior, GD with adaptive step sizes can successfully overcome the optimization challenge early in training, i.e., the iterate is sufficiently away from the spurious local optima at the end of Stage I. Meanwhile, (9) actually demonstrates that the algorithm enters the basin of attraction of the global optimum, and we next show the convergence of GD.

### 3.2 Stage II: Converge to the Global Optimum

Recall that in the previous stage, we use a conservative step size $\eta_w$ to avoid being trapped by the spurious local optimum. However, the small step size $\eta_w$ slows down the convergence of $w$ in the basin of attraction of the global optimum. Now we choose larger step sizes to accelerate the convergence. The following theorem shows that, after Stage I, we can use a larger $\eta_w$, while the results in Theorem 4 still hold, i.e., the iterate stays in the basin of attraction of $(w^*, a^*)$.

**Theorem 10.** *We restart the counter of time. Suppose $m \leq a_0^\top a^* \leq M$, and $\phi_0 \leq \frac{5}{12}\pi$. We choose $\eta_w \leq \frac{m}{M^2} = \widetilde{O}(\frac{1}{k^2})$ and $\eta_a < \frac{2\pi}{k+\pi-1}$. Then for all $t > 0$, we have*

$$\phi_t \leq 5\pi/12 \quad \text{and} \quad 0 \leq m \leq a_t^\top a^* \leq M.$$

*Proof Sketch.* To prove the first argument, we need the partial dissipativity of $\nabla_w \mathcal{L}$.

**Lemma 11.** *For any $m > 0$, $\nabla_w \mathcal{L}$ satisfies*

$$\langle -\nabla_w \mathcal{L}(w, a), w^* - w \rangle \geq \frac{m}{8} \|w - w^*\|_2^2,$$

*for any $(w, a) \in \mathcal{K}_m$, where*

$$\mathcal{K}_m = \left\{ (w, a) \mid a^\top a^* \geq m, \ (w + \mathbb{1}/\sqrt{p})^\top v^* \geq 0, \ \|w + \mathbb{1}/\sqrt{p}\|_2 = 1 \right\}.$$

This condition ensures that when $a^\top a^*$ is positive, $w$ always makes positive progress towards $w^*$, or equivalently $\phi_t$ decreasing. We need not worry about $\phi_t$ getting obtuse, and thus a larger step size $\eta_w$ can be adopted. The second argument can be proved following similar lines to Lemma 9. Please see Appendix B.3.2 for more details. $\qquad\square$

Now we are ready to show the convergence of our GD algorithm. Note that Theorem 10 and Lemma 11 together show that the iterate stays in the partially dissipative region $\mathcal{K}_w$, which leads to the convergence of $w$. Moreover, as shown in the following lemma, when $w$ is accurate enough, the partial gradient with respect to $a$ enjoys partial dissipativity.

**Lemma 12.** *For any $\delta > 0$, $\nabla_a \mathcal{L}$ satisfies*

$$\langle -\nabla_a \mathcal{L}(w, a), a^* - a \rangle \geq \frac{\pi - 1}{2\pi} \|a - a^*\|_2^2 - \frac{1}{5}\delta,$$

*for any $(w, a) \in \mathcal{A}_{m,M,\delta}$, where*

$$\mathcal{A}_{m,M,\delta} = \left\{ (w, a) \mid a^\top a^* \in [m, M], \ \|w - w^*\|_2^2 \leq \delta, \ \|w + \mathbb{1}/\sqrt{p}\|_2 = 1 \right\}.$$

As a direct result, $a$ converges to $a^*$. The next theorem formalize the above discussion.

**Theorem 13** (Convergence). *Suppose $\frac{1}{5}\|a^*\|_2^2 = m \leq a_t^\top a^* \leq M = 3\|a^*\|_2^2 + 2 \left( \mathbb{1}^\top a^* \right)^2$ hold for all $t > 0$. For any $\delta > 0$, choose $\eta_a = \eta_w = \eta = \min \left\{ \frac{m}{2M^2}, \frac{5\pi^2}{4(k+\pi-1)^2} \right\} = \widetilde{O}(\frac{1}{k^2})$, then we have*

$$\|w_t - w^*\|_2^2 \leq \delta \ \text{and} \ \|a_t - a^*\|_2^2 \leq 5\delta$$

*for any $t \geq T_2 = \widetilde{O}(\frac{1}{\eta} \log \frac{1}{\delta})$.*

*Proof Sketch.* The detailed proof is provided in Appendix B.4. Our proof relies on the partial dissipativity of $\nabla_w \mathcal{L}$ (Lemma 11) and that of $\nabla_a \mathcal{L}$ (Lemma 12).

Note that the partial dissipative region $\mathcal{A}_{m,M,\delta}$, depends on the precision of $w$. Thus, we first show the convergence of $w$.

**Lemma 14** (Convergence of $w_t$). *Suppose $\frac{1}{5}\|a^*\|_2^2 = m \leq a_t^\top a^* \leq M = 3\|a^*\|_2^2 + 4 \left( \mathbb{1}^\top a^* \right)^2$ hold for all $t > 0$. For any $\delta > 0$, choose $\eta \leq \frac{m}{2M^2} = \widetilde{O}(\frac{1}{k^2})$, then we have*

$$\|w_t - w^*\|_2^2 \leq \delta$$

*for any $t \geq \tau_{21} = \frac{4}{m\eta} \log \frac{4}{\delta} = \widetilde{O}(\frac{1}{\eta} \log \frac{1}{\delta})$.*

Lemma 14 implies that after $\tau_{21}$ iterations, the algorithm enters $\mathcal{A}_{m,M,\delta}$. Then we show the convergence property of $a$ in next lemma.

**Lemma 15** (Convergence of $a_t$). *Suppose $m \leq a_t^\top a^* \leq M$ and $\|w_t - w^*\|_2^2 \leq \delta$ holds for all $t$. We choose $\eta \leq \frac{5\pi^2}{4(k+\pi-1)^2} = O(\frac{1}{k^2})$. Then for all $t \geq \tau_{22} = \frac{4}{\eta} \log \frac{\|a_0 - a^*\|_2^2}{\delta} = \widetilde{O}(\frac{1}{\eta} \log \frac{1}{\delta})$, we have*

$$\|a_t - a^*\|_2^2 \leq 5\delta.$$

Combine the above two lemmas together, take $T_2 = \tau_{21} + \tau_{22}$, and we complete the proof. $\qquad\square$

Theorem 13 shows that with larger $\eta_w$ than in Stage I, GD converges to the global optimum in polynomial time. Compared to the convergence with constant probability for CNN (Du et al., 2017), Assumption 1 assures convergence even under arbitrary initialization of $a$. This partially justifies the importance of shortcut in ResNet.

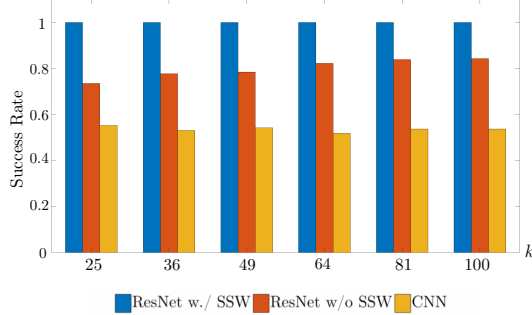

Figure 4: Success rates of converging to the global optimum for GD training ResNet with and without SSW and CNN with varying $k$ and and $p = 8$.

## 4 Numerical Experiment

We present numerical experiments to illustrate the convergence of the GD algorithm. We first demonstrate that with the shortcut prior, our choice of step sizes and the initialization guarantee the convergence of GD. We consider the training of a two-layer non-overlapping convolutional ResNet by solving (7). Specifically, we set $p = 8$ and $k \in \{16, 25, 36, 49, 64, 81, 100\}$. The teacher network is set with parameters $a^*$ satisfying $\mathbb{1}^\top a^* = \frac{1}{4}\|a^*\|_2^2$, and $v^*$ satisfying $v_1^* = \cos(7\pi/10)$, $v_2^* = \sin(7\pi/10)$, and $v_j^* = 0$ for $j = 3, \ldots, p$.[3] More detailed experimental setting is provided in Appendix C. We initialize with $w_0 = 0$ and $a_0$ uniformly distributed over $\mathbb{B}_0(|\mathbb{1}^\top a^*|/\sqrt{k})$. We adopt the following learning rate scheme with Step Size Warmup (SSW) suggested in Section 3: We first choose step sizes $\eta_a = 1/k^2$ and $\eta_w = \eta_a^2$, and run for 1000 iterations. Then, we choose $\eta_a = \eta_w = 1/k^2$. We also consider learning the same teacher network using step sizes $\eta_w = \eta_a = 1/k^2$ throughout, i.e., without step size warmup.

We further demonstrate learning the aforementioned teacher network using a student network of the same architecture. Specifically, we keep $a^*, v^*$ unchanged. We use the GD in Du et al. (2017) with step size $\eta = 0.1$, and initialize $v_0$ uniformly distributed over the unit sphere and $a$ uniformly distributed over $\mathbb{B}_0(|\mathbb{1}^\top a^*|/\sqrt{k})$.

For each combination of $k$ and $a^*$, we repeat 5000 simulations for aforementioned three settings, and report the success rate of converging to the global optimum in Table 1 and Figure 4. As can be seen, our GD on ResNet can avoid the spurious local optimum, and converge to the global optimum in all 5000 simulations. However, GD without SSW can be trapped in the spurious local optimum. The failure probability diminishes as the dimension increase. Learning the teacher network using a two-layer CNN student network (Du et al., 2017) can also be trapped in the spurious local optimum.

Table 1: *Success rates of converging to the global optimum for GD training ResNet with and without SSW and CNN with varying $k$ and and $p = 8$.*

| $k$ | 16 | 25 | 36 | 49 | 64 | 81 | 100 |
|---|---|---|---|---|---|---|---|
| ResNet w/ SSW | 1.0000 | 1.0000 | 1.0000 | 1.0000 | 1.0000 | 1.0000 | 1.0000 |
| ResNet w/o SSW | 0.7042 | 0.7354 | 0.7776 | 0.7848 | 0.8220 | 0.8388 | 0.8426 |
| CNN | 0.5348 | 0.5528 | 0.5312 | 0.5426 | 0.5192 | 0.5368 | 0.5374 |

We then demonstrate the algorithmic behavior of our GD. We set $k = 25$ for the teacher network, and other parameters the same as in the previous experiment. We initialize $w_0 = 0$ and $a_0 \in \mathbb{B}_0(|\mathbb{1}^\top a^*|/\sqrt{k})$. We start with $\eta_a = 1/k^2$ and $\eta_w = \eta_a^2$. After 1000 iterations, we set the step sizes $\eta_a = \eta_w = 1/k^2$. The algorithm is terminated when $\|a_t - a^*\|_2^2 + \|w_t - w^*\|_2^2 \leq 10^{-6}$. We also demonstrate the GD algorithm without SSW at the same initialization. The step sizes are $\eta_a = \eta_w = 1/k^2$ throughout the training.

One solution path of GD with SSW is shown in the first row of Figure 5. As can be seen, the algorithm has a phase transition. In the first stage, we observe that $w_t$ makes very slow progress due to the

small step size $\eta_w$, while $a_t^\top a^*$ gradually increases. This implies the algorithm avoids being attracted by the spurious local optimum. In the second stage, $w_t$ and $a_t$ both continuously evolve towards the global optimum.

The second row of Figure 5 illustrates the trajectory of GD without SSW being trapped by the spurious local optimum. Specifically, $(w_t, a_t)$ converges to $(\bar{w}, \bar{a})$ as we observe that $\phi_t$ converges to $\pi$, and $\|w_t - w^*\|_2^2$ converges to $4\|v^*\|_2^2$.

# 5 Discussions

**Deep ResNet.** Our two-layer network model is largely simplified compared with deep and wide ResNets in practice, where the role of the shortcut connection is more complicated. It is worth mentioning that the empirical results in Veit et al. (2016) show that ResNet can be viewed as an ensemble of smaller networks, and most of the smaller networks are shallow due to the shortcut connection. They also suggest that the training is dominated by the shallow smaller networks. We are interested in investigating whether these shallow smaller networks possesses similar benign properties to ease the training as our two-layer model.

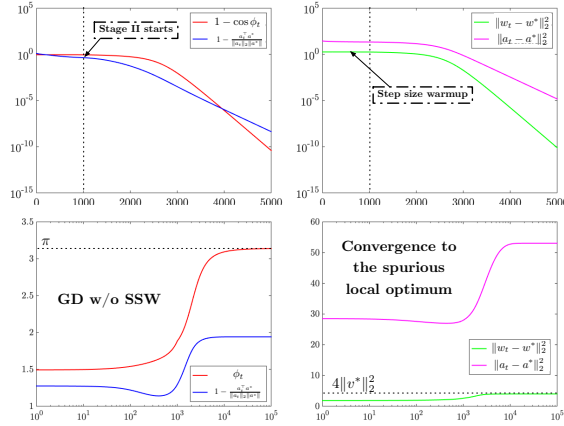

Figure 5: Algorithmic behavior of GD on ResNet. The horizontal axis corresponds to the number of iterations.

Moreover, our student network and the teacher network have the same degree of freedom. We have not considered deeper and wider student networks. It is also worth an investigation that what is the role of shortcut connections in deeper and wider networks.

**From GD to SGD.** A straightforward extension is to investigate the convergence of SGD with mini-batch. We remark that when the batch size is large, the effect of the noise on gradient is limited and SGD mimics the behavior of GD. When the batch size is small, the noise on gradient plays a significant role in training, which is technically more challenging.

**Related Work.** Li and Yuan (2017) study ResNet-type two-layer neural networks with the output weight known ($a = \mathbb{1}$), which is equivalent to assuming $a_t^\top a^* > 0$ for all $t$ in our analysis. Thus, their analysis does not have Stage I ($a_0^\top a^* < 0$). Moreover, since they do not need to optimize $a$, they only need to handle the partial dissipativity of $\nabla\mathcal{L}_w$ with $\delta = 0$ (one-point convexity). In our analysis, however, we also need to handle the the partial dissipativity of $\nabla\mathcal{L}_a$ with $\delta \neq 0$, which makes our proof more involved.

**Initialization.** Our analysis shows that GD converges to the global optimum, when $w$ is initialized at zero. Empirical results in Li et al. (2016) and Zhang et al. (2019) also suggest that deep ResNet works well, when the weights are simply initialized at zero or using the Fixup initialization. We are interested in building a connection between training a two-layer ResNet and its deep counterpart.

**Step Size Warmup.** Our choice of step size $\eta_w$ is related to the learning rate warmup and layerwise learning rate in the existing literature. Specifically, Goyal et al. (2017) presents an effective learning rate scheme for training ResNet on ImageNet for less than 1 hour. They start with a small step size, gradually increase (linear scale) it, and finally shrink it for convergence. Our analysis suggests that in the first stage, we need smaller $\eta_w$ to avoid being attracted by the spurious local optimum. This is essentially consistent with Goyal et al. (2017). Note that we are considering GD (no noise), hence, we do not need to shrink the step size in the final stage. While Goyal et al. (2017) need to shrink the step size to control the noise in SGD. Similar learning rate schemes are proposed by Smith (2017).

On the other hand, we incorporate the shortcut prior, and adopt a smaller step size for the inner layer, and a larger step size for the outer layer. Such a choice of step size is shown to be helpful in both deep learning and transfer learning (Singh et al., 2015; Howard and Ruder, 2018), where it is referred to as differential learning rates or discriminative fine-tuning. It is interesting to build a connection between our theoretical discoveries and these empirical observations.

## Footnotes

[2]The probability is bounded between $1/4$ and $3/4$. Numerical experiments show that this probability can be as bad as $1/2$ with the worst configuration of $a, v$.

[3]$v^*$ essentially satisfies $\angle(v^*, \mathbb{1}/\sqrt{p}) = 0.45\pi$.

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
