[Supplementary Material]

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

# Supplementary Material for Understanding the Importance of Shortcut Connections in ResNet

## A  Preliminaries

We first provide the explicit forms of the loss function and its gradients with respect to $w$ and $a$.

**Proposition 16.** *Let $\phi = \angle(\mathbb{1}/\sqrt{p} + w, v^*)$. When $\|\mathbb{1}/\sqrt{p} + w\|_2 = 1$, the loss function $\mathcal{L}(w, a)$ and the gradient w.r.t $(w, a)$, i.e., $\nabla_a \mathcal{L}(w, a)$ and $\nabla_w \mathcal{L}(w, a)$ have the following analytic forms.*

$$\mathcal{L}(w, a) = \frac{1}{2}\Big[\frac{(\pi - 1)}{2\pi}\|a^*\|_2^2 + \frac{(\pi - 1)}{2\pi}\|a\|_2^2 - \frac{1}{\pi}(g(\phi) - 1)a^\top a^*$$
$$+ \frac{1}{2\pi}(\mathbb{1}^\top a^*)^2 + \frac{1}{2\pi}(\mathbb{1}^\top a)^2 - \frac{1}{\pi}\mathbb{1}^\top a^* a^\top \mathbb{1}\Big],$$

$$\nabla_a \mathcal{L}(a, w) = \frac{1}{2\pi}(\mathbb{1}\mathbb{1}^\top + (\pi - 1))a - \frac{1}{2\pi}(\mathbb{1}\mathbb{1}^\top + (g(\phi) - 1))a^*,$$

$$\nabla_w \mathcal{L}(w, a) = -\frac{a^\top a^* (\pi - \phi)}{2\pi}\left(I - (\mathbb{1}/\sqrt{p} + w)(\mathbb{1}/\sqrt{p} + w)^\top\right)v^*,$$

*where $g(\phi) = (\pi - \phi)\cos(\phi) + \sin(\phi)$.*

This proposition is a simple extension of Theorem 3.1 in Du et al. (2017). Here, we omit the proof. For notational simplicity, we denote $v_t = \mathbb{1}/\sqrt{p} + w_t$ in the future proof.

## B  Proof of Theoretical Results

### B.1  Proof of Proposition B.1

*Proof.* Recall that Du et al. (2017) proves that $(\bar{v}, \bar{a}) = (-v^*, (\mathbb{1}\mathbb{1}^\top + (\pi - 1)I)^{-1}(\mathbb{1}\mathbb{1}^\top - I)a^*)$ is the spurious local optimum of the CNN counterpart to our ResNet. Substitute $\bar{v}$ by $\frac{\mathbb{1}/\sqrt{p}+w}{\|\mathbb{1}/\sqrt{p}+w\|_2}$ and we prove the result. $\qquad\square$

### B.2  Proof of Theorem 4

#### B.2.1  Proof of Lemma 5

*Proof.* By simple manipulication, we know that the initialization of $a$ satisfies $-2(\mathbb{1}^\top a^*)^2 \leq \mathbb{1}^\top a^* \mathbb{1}^\top a_0 - (\mathbb{1}^\top a^*)^2 \leq 0$. We first prove the right side of the inequality. Expand $a_t$ as $a_{t-1} - \eta_a \nabla_a \mathcal{L}(w_{t-1}, a_{t-1})$, and we have

$$\mathbb{1}^\top a^* \mathbb{1}^\top a_t = \left(1 - \frac{\eta_a (k + \pi - 1)}{2\pi}\right)\mathbb{1}^\top a^* \mathbb{1}^\top a_{t-1} + \frac{\eta_a (k + g(\phi_{t-1}) - 1)}{2\pi}(\mathbb{1}^\top a^*)^2$$
$$\leq \left(1 - \frac{\eta_a (k + \pi - 1)}{2\pi}\right)\mathbb{1}^\top a^* \mathbb{1}^\top a_{t-1} + \frac{\eta_a (k + \pi - 1)}{2\pi}(\mathbb{1}^\top a^*)^2.$$

Subtract $(\mathbb{1}^\top a^*)^2$ from both sides, then we get

$$\mathbb{1}^\top a^* \mathbb{1}^\top a_t - (\mathbb{1}^\top a^*)^2 \leq \left(1 - \frac{\eta_a (k + \pi - 1)}{2\pi}\right)(\mathbb{1}^\top a^* \mathbb{1}^\top a_{t-1} - (\mathbb{1}^\top a^*)^2)$$
$$\leq \left(1 - \frac{\eta_a (k + \pi - 1)}{2\pi}\right)^t (\mathbb{1}^\top a^* \mathbb{1}^\top a_0 - (\mathbb{1}^\top a^*)^2) \leq 0,$$

for any $t \geq 1$. The right side inequality is proved.

The proof of the left side follows similar lines. Since $g(\phi) \geq 0$, we have

$$\mathbb{1}^\top a^* \mathbb{1}^\top a_t = \left(1 - \frac{\eta_a\left(k + \pi - 1\right)}{2\pi}\right)\mathbb{1}^\top a^* \mathbb{1}^\top a_{t-1} + \frac{\eta_a\left(k + g(\phi_{t-1}) - 1\right)}{2\pi}\left(\mathbb{1}^\top a^*\right)^2$$

$$\geq \left(1 - \frac{\eta_a\left(k + \pi - 1\right)}{2\pi}\right)\mathbb{1}^\top a^* \mathbb{1}^\top a_{t-1} + \eta_a \frac{k-1}{2\pi}\left(\mathbb{1}^\top a^*\right)^2,$$

which is equivalent to the following inequality.

$$\mathbb{1}^\top a^* \mathbb{1}^\top a_t - \left(\mathbb{1}^\top a^*\right)^2 \geq \left(1 - \frac{\eta_a\left(k + \pi - 1\right)}{2\pi}\right)\left(\mathbb{1}^\top a^* \mathbb{1}^\top a_{t-1} - \left(\mathbb{1}^\top a^*\right)^2\right) - \frac{\eta_a}{2}\left(\mathbb{1}^\top a^*\right)^2.$$

$$\geq \left(1 - \frac{\eta_a\left(k + \pi - 1\right)}{2\pi}\right)^t\left(\mathbb{1}^\top a^* \mathbb{1}^\top a_0 - \left(\mathbb{1}^\top a^*\right)^2\right) - \frac{1}{1 - \left(1 - \frac{\eta_a\left(k + \pi - 1\right)}{2\pi}\right)}\frac{\eta_a}{2}\left(\mathbb{1}^\top a^*\right)^2$$

$$\geq \left(1 - \frac{\eta_a\left(k + \pi - 1\right)}{2\pi}\right)^t\left(\mathbb{1}^\top a^* \mathbb{1}^\top a_0 - \left(\mathbb{1}^\top a^*\right)^2\right) - \frac{\pi}{k + \pi - 1}\left(\mathbb{1}^\top a^*\right)^2$$

$$\geq \left(1 - \frac{\eta_a\left(k + \pi - 1\right)}{2\pi}\right)^t\left(-2\left(\mathbb{1}^\top a^*\right)^2\right) - \frac{\pi}{k + \pi - 1}\left(\mathbb{1}^\top a^*\right)^2$$

$$\geq -3\left(\mathbb{1}^\top a^*\right)^2.$$

Then we prove the lemma. $\qquad\qquad\square$

### B.2.2 Proof of Lemma 6

*Proof.* For each iteration, the distance of $w_t$ moving towards $\bar{w}$ is upper bounded by the product of the step size $\eta_w$ and the norm of the gradient $\nabla_w \mathcal{L}(w, a)$. We first bound the norm of the gradient. From the analytic form of $\nabla_w \mathcal{L}(w, a)$, we need to bound $a^\top a^*$. We first have the following lower bound.

$$a_{t+1} a^* = \left(1 - \frac{\eta_a\left(\pi - 1\right)}{2\pi}\right)a_t^\top a^* + \frac{\eta_a\left(g(\phi_t) - 1\right)}{2\pi}\|a^*\|_2^2 + \frac{\eta_a}{2\pi}\left(\left(\mathbb{1}^\top a^*\right)^2 - \mathbb{1}^\top a^* \mathbb{1}^\top a_t\right)$$

$$\geq \left(1 - \frac{\eta_a\left(\pi - 1\right)}{2\pi}\right)a_t^\top a^* - \eta_a \frac{2}{\pi}\|a^*\|_2^2,$$

which is equivalent to

$$a_{t+1} a^* + \frac{4}{\pi - 1}\|a^*\|_2^2 \geq \left(1 - \frac{\eta_a\left(\pi - 1\right)}{2\pi}\right)\left(a_t^\top a^* + \frac{4}{\pi - 1}\|a^*\|_2^2\right)$$

$$\geq \left(1 - \frac{\eta_a\left(\pi - 1\right)}{2\pi}\right)^{t+1}\left(a_0^\top a^* + \frac{4}{\pi - 1}\|a^*\|_2^2\right).$$

Since $a_0^\top a^* \geq -\|a^*\|_2^2$, we have $a_0^\top a^* + \frac{4}{\pi-1}\|a^*\|_2^2 \geq 0$. Thus, when $\eta_a < \frac{2\pi}{\pi-1}$,

$$a_{t+1} a^* \geq -\frac{4}{\pi - 1}\|a^*\|_2^2 \geq -2\|a^*\|_2^2.$$

When $a_{t+1} a^* < 2\|a^*\|_2^2$, the following inequality holds true.

$$\|\nabla_w L(w_t, a_t)\|_2^2 = \frac{\left(a_t^\top a^*\right)^2\left(\pi - \phi_t\right)^2}{4\pi^2}v^{*\top}\left(I - v_t v_t^\top\right)v^*$$

$$\leq \|a^*\|_2^4(I - v_t^\top v^*)(I + v_t^\top v^*) \leq \|a^*\|_2^4\|v_t - v^*\|_2^2.$$

We next prove that when $\eta_w$ is small enough, $\phi_t < \pi/2$ holds for all $t \leq T = O(1/\eta_a^2)$. We first have the following inequality.

$$1 \leq \|\widetilde{v}_{t+1}\|_2 = \sqrt{\|v_t\|_2^2 + \|\eta_w \nabla_w L(w_t, a_t)\|_2^2} \leq 1 + \|\eta_w \nabla_w L(w_t, a_t)\|_2.$$

Under Assumption 1, we know that $\phi_0 < \pi/3$. Then we can bound the norm of the difference between iterates $w_{t+1}$ and $w^*$ as follows.

$$\|v_{t+1} - v^*\|_2 = \|\widetilde{v}_{t+1}/\|\widetilde{v}_{t+1}\|_2 - v^*\|_2 \leq \frac{1}{\|\widetilde{v}_{t+1}\|_2}\|\widetilde{v}_{t+1} - v^*\|_2 + 1 - \frac{1}{\|\widetilde{v}_{t+1}\|_2}$$

$$\leq \|\widetilde{v}_{t+1} - v^*\|_2 + 1 - \frac{1}{1 + \|\eta_w \nabla_w L(w_t, a_t)\|_2}.$$

427 Plug in the upper bound of the norm of $\nabla_w L(w_t, a_t)$, and we obtain

$$\|v_{t+1} - v^*\|_2 \leq \|\widetilde{v}_{t+1} - v^*\|_2 + 1 - \frac{1}{1 + \eta_w\|a^*\|_2^2\|v_t - v^*\|_2}$$

$$= \|v_t - v^* - \eta_w\nabla_w\mathcal{L}(a_t, w_t)\|_2^2 + \frac{\eta_w\|a^*\|_2^2\|v_t - v^*\|_2}{1 + \eta_w\|a^*\|_2^2\|v_t - v^*\|_2}$$

$$\leq \|v_t - v^*\|_2 + \eta_w\|\nabla_w\mathcal{L}(a_t, w_t)\|_2 + \eta_w\|a^*\|_2^2\|v_t - v^*\|_2$$

$$\leq \|v_t - v^*\|_2 + \eta_w\|a^*\|_2^2\|v_t - v^*\|_2 + \eta_w\|a^*\|_2^2\|v_t - v^*\|_2$$

$$= (1 + 2\eta_w\|a^*\|_2^2)\|v_t - v^*\|_2 \leq (1 + 2\eta_w\|a^*\|_2^2)^t\|v_0 - v^*\|_2$$

$$\leq \exp(2t\eta_w\|a^*\|_2^2)\|v_0 - v^*\|_2$$

$$\leq \exp(2t\eta_w\|a^*\|_2^2) \leq 2 - 2\cos\left(\frac{5}{12}\pi\right),$$

428 for all $t \leq T = O(1/\eta_a^2)$, when $\eta_w = C_1\|a^*\|_2^2\eta_a^2 = \widetilde{O}(\eta_a^2)$ for some constant $C_1 > 0$. Thus
429 $\phi_t \leq \frac{5}{12}\pi$ for all $t \leq T = O(1/\eta_a^2)$. $\square$

### B.2.3   Proof of Lemma 7

430

*Proof.* For any $C_3 \in (0, 1)$, if we have $a^\top a^* \leq C_3\|a^*\|_2^2$, the norm of the difference between $a$ and $a^*$ satisfies the following inequality.

$$\|a - a^*\|_2^2 \geq (1 - 2C_3)\|a^*\|_2^2.$$

431 Let $C_2 = g(\frac{5}{12}\pi) - 1 = 0.4402$. SInce $\phi \leq \frac{5}{12}\pi$, and $g$ is strictly decreasing, we know that
432 $g(\phi) \geq C_2$. Using the above two inequalities, we can lower bound the inner product between the
433 negative gradient and the difference between $a$ and $a^*$ as follows.

$$\langle -\nabla_a L(w + \xi, a + \epsilon), a^* - a\rangle = \frac{1}{2\pi}\left(\mathbf{1}^\top a - \mathbf{1}^\top a^*\right)^2 + \frac{1}{2\pi}\left((\pi - 1)a - (g(\phi) - 1)a^*\right)^\top(a - a^*)$$

$$= \frac{1}{2\pi}\left(\mathbf{1}^\top a - \mathbf{1}^\top a^*\right)^2 + \frac{1}{2\pi}(\pi - g(\phi))a^\top(a - a^*) + \frac{g(\phi) - 1}{2\pi}\|a - a^*\|_2$$

$$\geq -\frac{1}{2\pi}(\pi - g(\phi))a^\top a^* + \frac{g(\phi) - 1}{2\pi}\|a - a^*\|_2^2$$

$$\geq -\frac{1}{2\pi}(\pi - g(\phi))a^\top a^* + \frac{g(\phi) - 1}{4\pi}\|a - a^*\|_2^2 + \frac{g(\phi) - 1}{4\pi}\|a - a^*\|_2^2$$

$$\geq -\frac{C_3}{2}\|a^*\|_2^2 + \frac{C_2}{4\pi}(1 - 2C_3)\|a^*\|_2^2 + \frac{C_2}{4\pi}\|a - a^*\|_2^2$$

$$\geq \frac{C_2}{4\pi}\|a - a^*\|_2^2 \geq \frac{1}{10\pi}\|a - a^*\|_2^2,$$

434 when $C_3 \leq \frac{C_2}{2(C_2 + \pi)}$. Take $C_3 = \frac{1}{20}$, and we prove the result. $\square$

### B.2.4   Proof of Lemma 8

435

436 *Proof.* We prove the result by contradiction. Specifically, we show that if $a_t \in \mathcal{A}$ always holds,
437 there always exist some time $\tau$ such that $a_\tau \notin \mathcal{A}$, which is a contradiction. Formally, suppose
438 $\forall\tau \leq t, a_\tau \in \mathcal{A}$, then we have

$$\|a_{t+1} - a^*\|_2^2 = \|a_t - a^*\|_2^2 - 2\langle -\eta_a\mathbb{E}_{\xi,\epsilon}\nabla_a L(w_t, a_t), a^* - a_t\rangle \qquad (12)$$

$$+ \|\eta_a\nabla_a L(w_t, a_t)\|_2^2. \qquad (13)$$

439 The second term is lower bounded according to the partial dissipativity of $\nabla_a\mathcal{L}$. Thus, we only need
440 to bound the norm of the gradient.

$$\|\nabla_a L(w_t, a_t)\|_2^2 = \|\nabla_a L(w_t, a_t) - \nabla_a L(w^*, a^*)\|_2^2$$

$$= \|\frac{1}{2\pi}\left(\mathbf{1}\mathbf{1}^\top + (\pi - 1)I\right)(a_t - a^*) - \frac{g(\phi) - \pi}{2\pi}a^*\|_2^2$$

$$\leq \frac{1}{2\pi^2}\|\left(\mathbf{1}\mathbf{1}^\top + (\pi - 1)I\right)(a_t - a^*)\|_2^2 + \frac{1}{2}\|a^*\|_2^2$$

$$\leq \frac{(k + \pi - 1)^2}{\pi^2}\left(\|a_t - a^*\|_2^2\right) + \frac{1}{2}\|a^*\|_2^2.$$

441 Plug the above bound into (12), then we have

$$\|a_{t+1} - a^*\|_2^2 \leq \left(1 - \frac{\pi}{5}\eta_a + \eta_a^2 \frac{(k+\pi-1)^2}{\pi^2}\right)\|a_t - a^*\|_2^2 + \frac{\eta_a^2}{2}\|a^*\|_2^2$$

$$\leq (1 - \lambda_1)\|a_t - a^*\|_2^2 + b_1$$

$$\leq (1 - \lambda_1)^{t+1}\|a_0 - a^*\|_2^2 + \frac{b_1}{\lambda_1},$$

442 where $\lambda_1 = \frac{\pi}{5}\eta_a - \eta_a^2 \frac{(k+\pi-1)^2}{\pi^2}$ and $b_1 = \frac{\eta_a^2}{2}\|a^*\|_2^2$. When $\eta_a < \frac{\pi}{20(k+\pi-1)^2}$, we have $\frac{b_1}{\lambda_1} \leq \frac{\|a^*\|_2^2}{6}$.
443 Thus, after $\tau_{11} = O(\frac{1}{\eta_a})$ iterations, we have

$$\|a_{\tau_{11}} - a^*\|_2^2 < \frac{\|a^*\|_2^2}{4}.$$

On the other hand, $a_{\tau_{11}} \in \mathcal{A}$ implies that $\|a_{\tau_{11}} - a^*\|_2^2 \geq \frac{1}{4}\|a^*\|_2^2$. Thus, after $\tau_{11} = O(\frac{1}{\eta_a})$ iterations,
we have

$$\frac{1}{20}\|a^*\|_2^2 \leq a_t^\top a^* \text{ and } \|a_t - a^*/2\|_2^2 \leq \|a^*\|_2^2.$$

444 Moreover, $\|a_t - a^*/2\|_2^2 \leq \|a^*\|_2^2$ implies $a_t^\top a^* \leq 2\|a^*\|_2^2$, and we prove the lemma. $\square$

## 445 B.2.5 Proof of Lemma 9

446 *Proof.* We first prove the left side. Write $a_{t+1} = a_t - \eta_a \nabla_a \mathcal{L}(w, a)$ and we have

$$a_{t+1}^\top a^* = \left(1 - \frac{\eta_a(\pi-1)}{2\pi}\right)a_t^\top a^* + \frac{\eta_a(g(\phi_t)-1)}{2\pi}\|a^*\|_2^2 + \frac{\eta_a}{2\pi}\left((\mathbb{1}^\top a^*)^2 - \mathbb{1}^\top a^* \mathbb{1}^\top a_t\right)$$

$$\geq \left(1 - \frac{\eta_a(\pi-1)}{2\pi}\right)a_t^\top a^* + \eta_a \frac{C_2}{2\pi}\|a^*\|_2^2.$$

447 The last inequality holds since $g(\phi) \geq 1$ and $(\mathbb{1}^\top a^*)^2 - \mathbb{1}^\top a^* \mathbb{1}^\top a_t \geq 0$. Subtract $\frac{C_2}{\pi-1}\|a^*\|_2^2$ from
448 both sides and we have the following inequality

$$a_{t+1}^\top a^* - \frac{C_2}{\pi-1}\|a^*\|_2^2 \geq \left(1 - \frac{\eta_a(\pi-1)}{2\pi}\right)\left(a_t^\top a^* - \frac{C_2}{\pi-1}\|a^*\|_2^2\right)$$

$$\geq \left(1 - \frac{\eta_a(\pi-1)}{2\pi}\right)^t \left(a_0^\top a^* - \frac{C_2}{\pi-1}\|a^*\|_2^2\right).$$

449 Thus, when $t \geq \tau_{12} = \widetilde{O}(1/\eta_a) > 0$, we have $a_t^\top a^* \geq \frac{1}{5}\|a^*\|_2^2$.

450 For the right side, follows similar lines to the left side, we have

$$a_{t+1}^\top a^* = \left(1 - \frac{\eta_a(\pi-1)}{2\pi}\right)a_t^\top a^* + \frac{\eta_a(g(\phi_t)-1)}{2\pi}\|a^*\|_2^2 + \frac{\eta_a}{2\pi}\left((\mathbb{1}^\top a^*)^2 - \mathbb{1}^\top a^* \mathbb{1}^\top a_t\right)$$

$$\leq \left(1 - \frac{\eta_a(\pi-1)}{2\pi}\right)a_t^\top a^* + \eta_a \frac{\pi-1}{2\pi}\|a^*\|_2^2 + \eta_a \frac{3}{2\pi}(\mathbb{1}^\top a^*)^2$$

$$\leq \left(1 - \frac{\eta_a(\pi-1)}{2\pi}\right)^{t+1} a_0^\top a^* + \|a^*\|_2^2 + \frac{3}{\pi-1}(\mathbb{1}^\top a^*)^2.$$

451 Note that $a_0^\top a^* \leq 2\|a^*\|_2^2$. Thus, for all $t$, $a_{t+1}^\top a^* \leq 3\|a^*\|_2^2 + 2(\mathbb{1}^\top a^*)^2$. $\square$

## 452 B.3 proof of Theorem 10

## 453 B.3.1 Proof of Lemma 11

454 *Proof.* Note that $\|v_t\|_2 = \|v^*\|_2 = 1$, according to Proposition 16, the gradient with respect to $w$
455 can be rewritten as

$$\nabla_w \mathcal{L}(w_t, a_t) = -\frac{a_t^\top a_t^* (\pi - \phi_t)}{2\pi}\left(I - v_t v_t^\top\right)v^*.$$

456    Then we have the following inequality.

$$
\begin{aligned}
\langle -\nabla_w \mathcal{L}(w_t, a_t), w^* - w_t \rangle &= \langle -\nabla_w \mathcal{L}(w_t, a_t), v^* - v_t \rangle \\
&= \frac{a_t^\top a_t^* \, (\pi - \phi_t)}{2\pi} \left( 1 - (v_t^\top v^*)^2 \right) \\
&\geq \frac{m}{4} (1 - v_t^\top v^*) \\
&= \frac{m}{8} \| w - w^* \|_2^2 .
\end{aligned}
$$

457    $\qquad\qquad\qquad\qquad\qquad\qquad\qquad\qquad\qquad\qquad\qquad\qquad\qquad\qquad\qquad$ $\square$

### B.3.2    proof of Theorem 10

459    *Proof.* First, we bound the norm of the gradient as follows

$$
\| \nabla_w L(w, a) \|_2^2 = \frac{\left( a^\top a^* \right)^2 (\pi - \phi)^2}{4\pi^2} v^{*\top} \left( I - vv^\top \right) v^* \leq \frac{M^2}{4} (I - v^\top v^*)(I + v^\top v^*) \leq \frac{M^2}{4} \| v - v^* \|_2^2 .
$$

460    Next we show that $\| v_{t+1} - v^* \|_2^2 \leq \| \widetilde{v}_{t+1} - v^* \|_2^2$. We first have the following two inequalities.

$$
\| \widetilde{v}_{t+1} \|_2^2 = \| v_t \|_2^2 + \| \eta_w \nabla_w L(w_t, a_t) \|_2^2 \geq 1 .
$$

461

$$
\widetilde{v}_{t+1}^\top v^* = v_t^\top v^* + \eta_w \langle -\nabla_w \mathcal{L}(w_t + \xi, a_t + \epsilon), v^* - v_t \rangle \geq v_t^\top v^* > 0 .
$$

462    Thus, $0 < v_{t+1}^\top v^* \leq 1$. We then have

$$
\begin{aligned}
\| \widetilde{v}_{t+1} - v^* \|_2^2 &= 1 + \| \widetilde{v}_{t+1} \|_2^2 - 2 \| \widetilde{v}_{t+1} \|_2 v_{t+1}^\top v^* \\
&\geq 1 + 1 - 2 w_{t+1}^\top w^* = \| v_{t+1} - v^* \|_2^2 .
\end{aligned}
$$

463    Then the distance between $\widetilde{w}_{t+1}$ and $w^*$ is as follows.

$$
\begin{aligned}
\| v_{t+1} - v^* \|_2^2 \leq \| \widetilde{v}_{t+1} - v^* \|_2^2 &= \| w_t - \eta_w \nabla_w \mathcal{L}(a_t, w_t) - w^* \|_2^2 \\
&= \| v_t - v^* \|_2^2 + \| \eta_w \nabla_w \mathcal{L}(a_t, w_t) \|_2^2 - 2\langle -\nabla_w \mathcal{L}(w_t + \xi, a_t + \epsilon), v^* - v_t \rangle \\
&\leq (1 - \eta_w \frac{m}{4} + \eta_w^2 \frac{M^2}{4}) \| v_t - v^* \|_2^2 \leq \| v_t - v^* \|_2^2 ,
\end{aligned}
$$

464    when $\eta_w \leq \frac{m}{M^2}$. Thus, $\phi_t \leq \phi_0 \leq \frac{5}{12}\pi$. We prove the first part.

465    We then prove the second part. Using the same expansion as in Lemma 9, we get

$$
\begin{aligned}
a_{t+1}^\top a^* &= \left( 1 - \frac{\eta_a (\pi - 1)}{2\pi} \right) a_t^\top a^* + \frac{\eta_a (g(\phi_t) - 1)}{2\pi} \| a^* \|_2^2 + \frac{\eta_a}{2\pi} \left( (\mathbb{1}^\top a^*)^2 - \mathbb{1}^\top a^* \mathbb{1}^\top a_t \right) \\
&\geq \left( 1 - \frac{\eta_a (\pi - 1)}{2\pi} \right) a_t^\top a^* + \eta_a \frac{C_2}{2\pi} \| a^* \|_2^2 .
\end{aligned}
$$

466    Choose $\eta_a < \frac{2\pi}{\pi - 1}$, such that $1 - \frac{\eta_a(\pi - 1)}{2\pi} < 1$. If $m \leq a_t^\top a^* \leq \frac{C_2}{\pi - 1} \| a^* \|_2^2$, the following inequality
467    shows that $a_t^\top a^*$ increases over time.

$$
a_{t+1}^\top a^* \geq \left( 1 - \frac{\eta_a (\pi - 1)}{2\pi} \right) a_t^\top a^* + \eta_a \frac{C_2}{2\pi} \| a^* \|_2^2 \geq a_t^\top a^* \geq m .
$$

468    If $a_t^\top a^* \geq \frac{C_2}{\pi - 1} \| a^* \|_2^2$, we show that this inequality holds for all $t$.

$$
\begin{aligned}
a_{t+1}^\top a^* &\geq \left( 1 - \frac{\eta_a (\pi - 1)}{2\pi} \right) a_t^\top a^* + \eta_a \frac{C_2}{2\pi} \| a^* \|_2^2 , \\
&\geq \left( 1 - \frac{\eta_a (\pi - 1)}{2\pi} \right) \frac{C_2}{\pi - 1} \| a^* \|_2^2 + \eta_a \frac{C_2}{2\pi} \| a^* \|_2^2 = \frac{C_2}{\pi - 1} \| a^* \|_2^2
\end{aligned}
$$

469    Combine these two cases together, we have $a_{t+1}^\top a^* \geq \min\{m, \frac{C_2}{\pi - 1} \| a^* \|_2^2\} = m$. The other side
470    follows similar lines in Lemma 9. Here, we omit the proof. $\qquad\qquad\qquad\qquad\qquad$ $\square$

### B.4 Proof of Theorem 13

#### B.4.1 Proof of Lemma 12

*Proof.* Note that

$$\|w_t - w^*\|_2^2 \le \delta \iff \cos(\phi_t) \ge 1 - \frac{\delta}{2}.$$

Moreover, we can bound $g(\phi_t)$ as follows

$$\pi \ge g(\phi_t) = (\pi - \phi_t)\cos\phi_t + \sin\phi_t \ge \left(1 - \frac{\delta}{2}\right)\pi = \pi - \frac{\delta}{2}\pi.$$

Thus we have the partial dissipativity of $\nabla_a \mathcal{L}$.

$$\langle -\nabla_a L(w,a), a^* - a \rangle = \frac{1}{2\pi}\left(\mathbb{1}^\top a - \mathbb{1}^\top a^*\right)^2 + \frac{1}{2\pi}\left((\pi - 1)a - (g(\phi) - 1)a^*\right)^\top (a - a^*)$$

$$= \frac{1}{2\pi}\left(\mathbb{1}^\top a - \mathbb{1}^\top a^*\right)^2 + \frac{1}{2\pi}(\pi - g(\phi))a^{*\top}(a - a^*) + \frac{\pi - 1}{2\pi}\|a - a^*\|_2^2$$

$$\ge \frac{\pi - 1}{2\pi}\|a - a^*\|_2^2 - \delta/5.$$

$\square$

#### B.4.2 Proof of Lemma 14

*Proof.* First, we bound the norm of the gradient as follows

$$\|\nabla_w L(w,a)\|_2^2 = \frac{(a^\top a^*)^2 (\pi - \phi)^2}{4\pi^2} v^{*\top}\left(I - vv^\top\right)v^* \le \frac{M^2}{4}(I - v^\top v^*)(I + v^\top v^*) \le \frac{M^2}{4}\|v - v^*\|_2^2$$

We next show that $\|w_{t+1} - w^*\|_2^2 \le \|\widetilde{w}_{t+1} - w^*\|_2^2$. We first have the following inequality.

$$\|\widetilde{w}_{t+1}\|_2^2 = \|w_t\|_2^2 + \|\eta\nabla_w L(w_t, a_t)\|_2^2 \ge 1.$$

Since we have $w_{t+1}^\top w^* \le 1$, we show that $\|\widetilde{v}_{t+1} - v^*\|_2^2 \le \|v_{t+1} - v^*\|_2^2$.

$$\|\widetilde{v}_{t+1} - v^*\|_2^2 = 1 + \|\widetilde{w}_{t+1}\|_2^2 - 2\|\widetilde{w}_{t+1}\|_2 w_{t+1}^\top w^*$$

$$\ge 1 + 1 - 2w_{t+1}^\top w^* = \|v_{t+1} - v^*\|_2^2.$$

Then the distance between $\widetilde{w}_{t+1}$ and $w^*$ is as follows.

$$\|v_{t+1} - v^*\|_2^2 \le \|\widetilde{v}_{t+1} - v^*\|_2^2 = \|w_t - \eta\nabla_w\mathcal{L}(a_t, w_t) - w^*\|_2^2$$

$$= \|w_t - w^*\|_2^2 + \|\eta\nabla_w\mathcal{L}(a_t, w_t)\|_2^2 - 2\langle -\nabla_w\mathcal{L}(w + \xi, a + \epsilon), w^* - w\rangle$$

$$\le (1 - \eta\frac{m}{4} + \eta^2\frac{M^2}{4})\|v_t - v^*\|_2^2.$$

So we have for any $t$,

$$\|v_t - v^*\|_2^2 \le (1 - \eta\frac{m}{4} + \eta^2\frac{M^2}{4})^t\|v_0 - v^*\|_2^2.$$

Thus, choose $\eta \le \frac{m}{2M^2} = \widetilde{O}(\frac{1}{k^2})$, and after $t \ge \tau_{21} = \frac{4}{m\eta}\log\frac{4}{\delta}$ iterations, we have

$$\|v_t - v^*\|_2^2 \le \delta,$$

which is equivalent to

$$\|w_t - w^*\|_2^2 \le \delta.$$

$\square$

#### B.4.3 Proof of Lemma 15

*Proof.* The proof follows similar lines to that of Lemma 14. By the partial dissipativity of $\mathcal{L}_a$, we

have

$$\|a_{t+1} - a^*\|_2^2 = \|a_t - a^*\|_2^2 - 2\langle -\eta\mathbb{E}_{\xi,\epsilon}\nabla_a L(w_t, a_t), a^* - a_t\rangle$$

$$+ \|\eta\nabla_a L(w_t, a_t)\|_2^2$$

$$\le \left(1 - \eta\frac{\pi - 1}{\pi} + \eta^2\frac{(k + \pi - 1)^2}{\pi^2}\right)\|a_t - a^*\|_2^2 + 2\eta^2\delta^2/25 + 2\eta\delta/5$$

$$\le (1 - \lambda_2)\|a_t - a^*\|_2^2 + \frac{4}{5}\eta\delta$$

$$\le (1 - \lambda_2)^{t+1}\|a_0 - a^*\|_2^2 + \frac{b_2}{\lambda_2}.$$

where $\lambda_2 = \eta\frac{\pi-1}{\pi} - \eta^2\frac{(k+\pi-1)^2}{\pi^2}$ and $b_2 = \frac{4}{5}\eta\delta$. Take $\eta \leq \frac{5\pi^2}{4(k+\pi-1)^2}$, and then $\lambda_2 \geq \frac{\eta}{4}$. When $t \geq \tau_{22} = \frac{4}{\eta}\log\frac{\|a_0-a^*\|_2^2}{\delta} = \widetilde{O}(\frac{1}{\eta}\log\frac{1}{\delta})$, we have

$$\|a_t - a^*\|_2^2 \leq 5\delta.$$

$\qquad\qquad\qquad\qquad\qquad\qquad\qquad\qquad\qquad\qquad\qquad\qquad\qquad\qquad\qquad\qquad\qquad\square$

## C  Experimental Settings

The output weight $a^*$ in the teacher network is chosen as in Table 2.

| $k$ | $(a^*)^\top$ |
|---|---|
| 16 | $[\underbrace{1,\ldots,1}_{9}, \underbrace{-1,\ldots,-1}_{7}]$ |
| 25 | $[\underbrace{1,\ldots,1}_{14}, \underbrace{-1,\ldots,-1}_{11}]$ |
| 36 | $[\underbrace{1,\ldots,1}_{19}, \underbrace{-1,\ldots,-1}_{16}, 0]$ |
| 49 | $[\underbrace{1,\ldots,1}_{26}, \underbrace{-1,\ldots,-1}_{22}, 0]$ |
| 64 | $[\underbrace{1,\ldots,1}_{34}, \underbrace{-1,\ldots,-1}_{30}]$ |
| 81 | $[\underbrace{1,\ldots,1}_{43}, \underbrace{-1,\ldots,-1}_{38}]$ |
| 100 | $[\underbrace{1,\ldots,1}_{52}, \underbrace{-1,\ldots,-1}_{47}, 0]$ |

Table 2: Output weight $a^*$.

The trajectories in Figure 3 are obtained with $a$ initialized at

$$
\begin{aligned}
a_0 =&[-0.1268, -0.1590, -0.1071, -0.1594, -0.4670, 0.1563, 0.1894, -0.2390, -0.0602, \\
&- 0.5047, 0.0325, -0.0886, 0.1514, -0.0883, -0.0243, 0.1198, -0.2805, 0.0024, \\
&- 0.0855, 0.0742, -0.0976, -0.1768, 0.1207, 0.0049, 0.1809].
\end{aligned}
$$