[Reviews · NeurIPS 2019]

Reviewer 1



Originality: The methods described in the paper have been studied vigorously over the past few years. The main contribution made by the paper is directly stated in the paper in the form of theorem 2. The paper highlight related work in the area that recognizes the problem statement i.e "GD gets trapped in spurious local minima in specific networks with shortcut connections " but is rather very brief in stating the line of work that aims to answer the question. Clarity: The methods and results described in the paper are extremely well written and easy to follow. The organization of different sections in the paper can be greatly improved. The related work is stated sporadically in various sections and there is only a small paragraph that is completely dedicated to related work. Such form of writing may be confusing for first-time readers to draw a connection between related work and methods presented in the paper. Additionally, It may be a better choice to split the last section into two sections future work and discussions. Significance: The main result of the paper is only applicable to small networks with average pooling shortcut connection, very specific weight initialization, specific step size etc. These results may not be directly extended to other shortcut connection based architectures. Thus, it may be helpful to have in-depth discussions on the extension of the presented work to other cases e.g would SGD show similar 2-stage convergence behavior? Will the convergence behavior extend to deeper networks? How to perform initialization, choose step size for deep networks?

Reviewer 2



The paper investigates the outcome of training a one hidden layer convolutional residual network architecture using gradient descent when input is sampled from standard Gaussian distribution. As a followup of a similar analysis of Du et al (2017) for CNNs, this paper shows for ResNets that there exists two fixed points to the teacher-student loss function (network architecture is same for both). While one is a global minimum, the other is a spurious fixed point. The authors then derive *sufficient* conditions on the parameter initialization and learning rates such that training happens in two phases: 1. first phase where the hidden layer weights (w) remain away from the spurious fixed point (due to sufficiently small learning rate) while the last layer weights (a) approach the optimal value and eventually enter the region where the inner product satisfies a'a* > 0. 2. second phase in which both parameters approach the global minimum such that the learning rate for w can be larger allowing faster convergence. I find this paper to be very interesting as it provides novel insights into the optimization process of ResNets even though in a very restricted setting. They also point out that the success probability of reaching the global minimum using their prescribed optimization hyper-parameters is 1 for ResNets while as shown in previous literature, this probability is only constant and less than 1 for CNNs. However I have the following concerns: 1. I do not see the condition w_0=0 being used in the proof of lemma 5. It seems unnecessary. 2. the conditions on initialization and learning rate derived in the paper are sufficient conditions for optimization to reach the global minimum. The conditions under which optimization may reach the spurious fixed point have not been discussed. This is why in table 1, the probability of optimization landing in the global minimum when not using step size warmup is between 0 and 1 and not 0. Some discussion around this would be useful. 3. From the proof strategy, it seems another learning rate schedule that would allow optimization with proposed initialization to reach the global minimum would be-- set the learning rate of w to 0 in the first phase while train a with the proposed learning rate until it converges to a*. Then train w while fixing the parameters a in the second phase until converges. In this way, the network can be trained in a layerwise manner. I encourage the authors to add a discussion around this. 4. The objective has a unique global minimum and spurious minimum due to the nature of the architecture and input distribution used in the analysis. What insights can we get from this analysis about the case where the fixed points are not unique (Eg. due to a different input distribution or architecture)? 5. In table 1, it can be seen that the success rate increases with increasing input dimensionality. While this is acknowledged in line 254, there is no discussion around it as to why this happens. Can authors provide an explanation? 6. In lines 304-311, the authors try to establish a relation between the strategy of learning rate warmup used in Goyal et al (2017) and the analysis provided by the authors on the learning rate schedule which suggests the use of small learning rate initially to avoid spurious fixed point. However, in the realistic case (eg. Goyal et al 2017 and many other papers that follow), using a large learning rate initially does not prevent training loss from reducing. It is the generalization that becomes worse. But this is a separate issue than the one the authors of this paper are addressing. So i would recommend the authors to revise their claim. 7. The authors again claim to establish a relationship between their analysis of using w=0 as initialization and the proposal in Fixup (Zhang et al 2019). Fixup proposes an initialization for ResNets without any normalization while the analysis in this paper is for a ResNet with normalization. So it is misleading to compare the two. Minor corrections: 1. Line 180: The dissipative region is for gradient of loss w.r.t. a, not w. 2. Line 230: It should be stage II, not I. 3. Line 263: It is first row, not first column. ##### Post Rebuttal: Thank you for the detailed explanations. Looking forward to the revised version with the recommenced changes.

Reviewer 3



Originality: The paper gives a new insight over why residual networks work in practice. It follows several prior research to analyze the theoretical part, but the understanding is new. Clarity: The paper is clearly written and well organized. Significance: In terms of understanding, the paper is somehow valuable. But why this understanding is only based on a two-layer non-overlapping convolutional neural networks? It assumes ||v||=1 and using ReLU as the nonlinear active function, what will it be if the assumptions are avoided? I am doubt the generalization of the understanding. For the convergence analysis, the paper provides some bounds for the convergency, but no compatibles are provided, it is hard to judge the tightness of the results.

[Author Response · NeurIPS 2019]

## Response to Reviews on "Towards Understanding the Importance of Shortcut Connections in Residual Networks"

We appreciate reviewers' valuable comments. We will correct typos and reply to comments in the following.

**To Reviewer #1**:

Our analysis can be extended to more general cases. We will add more discussions in the revision.

• Extension to SGD: Our analysis can be extended to mini-batch SGD when the batch size is large. We can show its convergence by applying tools from the super-martingale theory, however, the analysis is more involved.

• Extension to deep networks: As mentioned in Lines 71-77, the weights in well-trained deep ResNet have small magnitudes. Thus, we expect that the shortcut prior assumption generally holds true for deep ResNets, and consequently the shortcut connection can analogously ease the optimization as in our two-layer ResNet model. Moreover, the partial dissipativity condition (PD for short, Definition 3,) provides a potential outlet to analyze deep ResNet. We will provide an empirical verification of the partial dissipativity condition for deep ResNets in the revision.

• Initialization and step size: For the first layer $w$, simply initializing $w$ at 0 has been observed working well in deep ResNets (as mentioned in the previous paragraph, $w$ tends to have a small magnitude). Thus, we believe that initializing $w$ at origin can still work well in deep networks. For the second layer $a$, we use $O(1/\sqrt{k})$ type initialization. This coincides with common initialization techniques for deep networks (Glorot and Bengio, 2010; He et al., 2015; LeCun et al., 1998). Although deep networks undoubtedly need a more complex step size scheme, our analysis provides useful insights: For example, our choice of step sizes is consistent with the step size warmup scheme for deep ResNet (Goyal et al. 2017) as mentioned in Section 5.

**To Reviewer #2**:

**Comment 1:** Lemma 5 does not require $w_0 = 0$. We will remove it in the revision.

**Comment 2:** We briefly discuss the conditions for GD to be trapped in the spurious local optimum. Define the basins of attraction of the global optimum and spurious optimum as $\mathcal{R}_1 = \{(w,a)|a^\top a^* > 0, \phi = \angle(1/\sqrt{p} + w, 1/\sqrt{p} + w^*) < \pi/2\}$ and $\mathcal{R}_2 = \{(w,a)|a^\top a^* < 0, \phi = \angle(1/\sqrt{p} + w, 1/\sqrt{p} + w^*) > \pi/2\}$, respectively.

• Larger step size with $w = 0$: There exists a small constant $\epsilon > 0$, such that when $a_0^\top a^* < -\epsilon$, GD will be trapped in the spurious optimum. This is because $a_t^\top a^*$ takes a large amount of iterations to increase from negative to 0. Consequently, with a large step size, $w$ can move far away from $w^*$ before $a_t^\top a^*$ becomes nonnegative. This implies there exists $T$ such that $(w_T, a_T) \in \mathcal{R}_2$, i.e., GD will be trapped in the spurious local optimum. On the other hand, if $a_0^\top a^* > -\epsilon$, $a_t^\top a^*$ becomes positive in a few iterations, and $\phi < \pi/2$ still holds. GD stays in $\mathcal{R}_1$, and converges to the global optimum. Note that the constant $\epsilon$ depends on $w^*$ and is hard to characterize. We leave it for future investigation.

• Different initialization: If $(w_0, a_0)$ falls in $\mathcal{R}_2$, GD will get trapped in the spurious optimum.

**Comment 3:** We appreciate reviewer's suggestion. Training the network in a layer-wise manner (setting learning rate of $w$ to 0) is actually a special case considered in our analysis. We will add a discussion in the revision.

**Comment 4:** Our analysis applies to multiple spurious optima, as long as the partial dissipativity (PD) condition holds for the unique global optimum. For problems with multiple global optima, our analysis can still be applied if the following condition holds: there exists one global optimum such that the PD condition holds globally with respect to this optimum. In fact, we can empirically verify that PD condition holds globally with respect to the well trained model for some deep networks where multiple global optima exist. Thus, our analysis provides useful insights towards understanding multiple global optima cases. The theoretical analysis for multiple global optima cases needs a more detailed characterization of the landscape, which is technically difficult. We leave it for future investigation.

**Comment 5:** When we randomly initialize $a$ in the ball, we have $a_0^\top a^* = O_p(1/k)$, where $k$ is the dimension of $a$. This means $a_0$ has a larger chance to fall in the region where $a_0^\top a^* > -\epsilon$. As mentioned in the first item of our response to comment 2, GD initialized in this region will converge to the global optima. That is why we observe the increase of the success rate in our experiment. We further empirically observe that the success rate does not increase to 1 exponentially fast.

**Comment 6:** We show that using a small learning rate helps GD avoid the spurious optimum. In the two-layer ResNet, the spurious optimum yields bad generalization. We will make a clarification in the revision.

**Comment 7:** We thank the reviewer for this suggestion. We will revise our claim and make a clarification.

**To Reviewer #3**:

Our understanding of two-layer ResNet can provide useful insight towards understanding more general networks as we discussed in our response to Reviewer 1. The assumption $\|v\| = 1$ is used to stabilize the training, and ReLU is one of the most commonly used activation function. These assumptions eases our analysis but is not necessary. The key, partial dissipativity condition, is possible to hold for other networks. We remark that, even for this simple network, the analysis is already quite challenging.

We must emphasize that we are the FIRST to analyze the convergence of GD for two-layer ResNet, and there exists no other bound for comparison. We show that GD converge to the global optimum in polynomial time, but the degree of the polynomial may not be tight. In fact, the lower bound of GD is unknown and hard to characterize since GD can be trapped in the spurious optimum.

[Meta-Review · NeurIPS 2019]

The reviewers and AC carefully examined the author feedback and think it satisfactorily addresses the points raised in the reviews. We strongly urge the authors to incorporate this feedback in their manuscript.